# Tactics with Prebiotics for the Treatment of Metabolic Dysfunction-Associated Fatty Liver Disease via the Improvement of Mitophagy

**DOI:** 10.3390/ijms24065465

**Published:** 2023-03-13

**Authors:** Ai Tsuji, Sayuri Yoshikawa, Yuka Ikeda, Kurumi Taniguchi, Haruka Sawamura, Sae Morikawa, Moeka Nakashima, Tomoko Asai, Satoru Matsuda

**Affiliations:** Department of Food Science and Nutrition, Nara Women’s University, Kita-Uoya Nishimachi, Nara 630-8506, Japan

**Keywords:** prebiotics, NAFLD, MAFLD, mitophagy, ROS, mTOR, AMPK, PI3K/AKT signaling pathway

## Abstract

Mitophagy/autophagy plays a protective role in various forms of liver damage, by renovating cellular metabolism linking to sustain liver homeostasis. A characterized pathway for mitophagy is the phosphatase and tensin homolog (PTEN)-induced putative kinase 1 (PINK1)/Parkin-dependent signaling pathway. In particular, PINK1-mediated mitophagy could play an indispensable role in improving the metabolic dysfunction-associated fatty liver disease (MAFLD) which could precede to steatohepatitis (NASH), fibrosis, and hepatocellular carcinoma. In addition, the PI3K/AKT/mTOR pathway might regulate the various characteristics of cellular homeostasis including energy metabolism, cell proliferation, and/or cell protection. Therefore, targeting mitophagy with the alteration of PI3K/AKT/mTOR or PINK1/Parkin-dependent signaling to eliminate impaired mitochondria might be an attractive strategy for the treatment of MAFLD. In particular, the efficacy of prebiotics for the treatment of MAFLD has been suggested to be useful via the modulation of the PI3K/AKT/mTOR/AMPK pathway. Additionally, several edible phytochemicals could activate mitophagy for the improvement of mitochondrial damages, which could also be a promising option to treat MAFLD with providing liver protection. Here, the potential therapeutics with several phytochemicals has been discussed for the treatment of MAFLD. Tactics with a viewpoint of prospective probiotics might contribute to the development of therapeutic interventions.

## 1. Introduction

Prebiotics are additional effective interventions through dietary supplements of indigestible food ingredients that improve host health by selectively stimulating the growth and activity of bacteria in one or a small number of colonies. In other words, prebiotics could be defined as substrates that are utilized by beneficial bacteria providing a health benefit to the host [1]. The use of prebiotics may be a relevant option in human health [2]. In fact, previous studies have investigated the efficacy of prebiotics treatment for nonalcoholic fatty liver disease (NAFLD) [3]. For example, gellan gum had prebiotic activity, which is an anionic polysaccharide used as an additive in the food industry [4]. Gellan gum may promote liver health through the modulation of gut homeostasis [4]. In addition, low fiber consumption is associated with the prevalence of NAFLD [5]. Patients with NAFLD have a low daily fiber consumption [6]. Similarly, a high-fiber diet has been shown to be related to the recession of NAFLD [7]. Furthermore, pectin could modulate the gut microbiome, which is a soluble fiber found in different fruits and vegetables. It has been shown that pectin could protect liver from metabolic injuries caused by alcohol and/or fatty diet [6].

Mitochondrial activity is important for cell survival, proliferation, and/or differentiation [8]. In addition, many diseases such as age-related pathological conditions have been related to an altered mitochondrial function [9]. The importance of mitochondria has been emphasized in a variety of metabolic diseases including NAFLD [10]. NAFLD is a term covering the conditions characterized by excessive fat accumulation in the liver that are not triggered by alcohol intake. Excessive hepatic fat accumulation might be a hallmark of NAFLD [11]. NAFLD is the most common chronic liver disease affecting nearly 30% of the population [12]. Deskbound lifestyles, dietary changes, epidemic obesity, and type 2 diabetes have been identified as risk factors contributing to the increase in NAFLD. Patients with NAFLD are at an increased risk of developing steatohepatitis and/or nonalcoholic steatohepatitis (NASH) and fibrosis. NASH may ultimately lead to complications such as cirrhosis or liver failure. Nowadays, NASH has become among the top leading indications for liver transplantation [13]. Consequent complications of NASH or cirrhosis may include the development of hepatocellular carcinoma (HCC) [14]. It is a multisystem illness affecting numerous extrahepatic organ-diseases including type 2 diabetes and/or cardiovascular disease, all of which contribute to increased mortality in patients with NAFLD/NASH [15]. Recently, the redefinition of the NAFLD to metabolic-associated fatty liver disease (MAFLD) have been proposed to more accurately reflect the disease’s heterogeneity and pathogenesis [16]. Therefore, we hereafter used the term “MAFLD” instead of the term “NAFLD”. Mitochondrial dysfunction leads to the overproduction of ROS and hepatocyte apoptosis, which is closely related to MAFLD [17]. To slow down the progression of the pathology of MAFLD, the high oxidative stress and the mitochondrial activity of hepatocytes should be at least improved. Mitophagy, a selective degradation of mitochondria by autophagy, could selectively degrade damaged mitochondria to reverse mitochondrial dysfunction and/or maintain mitochondrial function. Mutations in the genes that encode essential factors for mitophagy may result in impaired mitophagy, which could lead to MAFLD [18]. Interestingly, it has been shown that the prebiotic administration of lactulose and/or trehalose could stimulate the autophagic pathway, which may result in the retrieval of learning in Alzheimer’s disease model mice [19]. The conserved pathway of autophagy might be required for preventing and/or counteracting pathogenic actions which could lead to the prevention of MAFLD as well as of neurodegenerative diseases. This review provides an overview of the molecular mechanism of mitochondrial dysfunction in MAFLD and discuss the potential therapeutics with prebiotics targeting autophagy for the treatment of MAFLD. Consequently, tactics with an emphasis of probiotics and its potential role in modulating metabolic liver disease might contribute to the development of treatment against the progression of MAFLD.

## 2. Dysfunction of Mitochondria Involved in the Pathogenesis of MAFLD

Mitochondria are dynamic organelles which can amend the abnormalities in cellular energy metabolism by regulating mitochondrial fusion, mitochondrial biogenesis, and the removal of injured mitochondria [20]. The most common explanation for mitochondrial dysfunction is hypoxia or the overproduction of ROS. In particular, oxidative stress with excess ROS caused by inflammation is intensely related to the development of MAFLD [21]. In addition, a high-fat and high-cholesterol diet could greatly raise the blood ROS level, suggesting that increased energy influx to the liver might be involved in ROS generation [21]. Excessive ROS correspondingly induces oxidative stress and apoptosis in cells [22]. Naturally in cells, ROS are generated in the mitochondrial respiratory chain. Accordingly, mitochondrial dysfunction could lead to hepatocyte cell death and/or apoptosis [23], which is intensely associated with MAFLD [24]. These oxidative stresses including oxygen radicals could be seized [25]. Indeed, this highlights the importance of healthy mitochondria for the maintenance of healthy hepatocyte functions. Under biological conditions, ROS could work as a regulatory mechanism for the homeostasis of cellular redox [26]. During oxidative phosphorylation in cellular respiration, mitochondria could produce by-product superoxide anions, which may be further changed into ROS [27]. The mitochondrial ROS involves hydrogen peroxide, superoxide, and hydroxyls, which could impair proteins, lipids, and/or DNAs resulting in mitochondrial dysfunction, then finally lead to cell death [28]. An increase in damaged mitochondria and/or the overproduction of ROS would augment each other, forming a malicious cycle. Mitochondrial ROS is also a regulator of cellular redox system linking to cellular metabolic stability [29]. Hence, impeding too much production of ROS might be reflected as an actual way to avoid oxidative damages to hepatocyte [30]. The antioxidants can initially safely be dismissed as ROS [31]. Additionally, there are various schemes to survive in ROS-induced oxidative stresses, such as catalase, glutathione peroxidase, and/or superoxide dismutase. In view of the essential role of mitochondria in cellular homeostasis, monitoring the quality of mitochondria might be important to evade the pathogenesis of MAFLD [32,33,34]. The functional deficiencies and/or changed morphology of mitochondria have been observed in the hepatocytes of MAFLD [35]. Likewise, it has been suggested that the dysfunction of mitochondria might also be a common feature of glycogen storage diseases [34]. Therefore, mitochondrial quality control is crucial to preventing the disease progression of liver diseases.

A mitochondrial quality control mechanism called mitophagy could selectively remove the damaged mitochondria to keep the mitochondrial function and/or sustain cellular homeostasis. Mitophagy is an autophagic response that specifically targets damaged mitochondria. It is well known that the clearance mechanism of injured mitochondria will be the potent therapeutic scheme for oxidative stresses [35]. In general, dysfunctional mitochondria could be removed through mitophagy. In this process, damaged mitochondria would be captured by autophagic membranes, then further delivered to lysosomes, in which the damaged mitochondria would be degraded. Some physiological errors in mitochondria, possibly due to oxidative stress, may initiate the mitophagy via the disruption of the mitochondrial membrane by ROS. (Figure 1) A reduction in mitochondrial quality has been suggested to play a major role in the development and/or progression of MAFLD [36,37]. Additionally, the imbalance of the mitophagy in hepatocytes could promote the development of MAFLD [38]. The balance between biogenesis and the degradation of mitochondria might be indispensable to keep healthy mitochondria and/or cellular homeostasis. Therefore, the modulation of the mitophagy may offer a promising way for the intervention of MAFLD.

## 3. Quality Control of Mitochondria via the Mitophagy Involved in MAFLD

Mitochondrial dysfunction might be reflected as a pathogenesis of metabolic disorders including MAFLD. Therefore, the mitophagy should separate the impaired mitochondria from healthy mitochondria, and selectively degrade the damaged mitochondria. To put it another way, mitophagy is the selective degradation of mitochondria by autophagy, which often occurs on defective mitochondria following damage and/or oxidative stress. In detail, for the molecular mechanisms of mitophagy, damaging mitochondria takes the lead in the accumulation of PTEN-induced kinase 1 (PINK1) in the outer membrane of mitochondria (OMM), where it employs E3 ubiquitin ligase Parkin which could start the elimination of damaged mitochondria by autophagosomes [39]. PINK1 is positioned in the outer mitochondrial membrane, but PINK1 cannot be detected in the OMM of healthy mitochondria. Thus, PINK1 and Parkin are recruited to the OMM for the elimination of impaired mitochondria [40], which in turn ubiquitinate the surface proteins of mitochondria [41] (Figure 1). PINK1 and Parkin are two well-known and important mediators regulating mitophagy in mammalian cells [42]. An activating kinase of PINK1 has microtubule affinity regulating kinase 2 (MARK2) [43]. Parkin could stimulate the ubiquitination of the mitochondrial fusion proteins mitofusin 1 (MFN1) and mitofusin 2 (MFN2) resulting in mitochondrial fission, which contributes to the separation of damaged mitochondria from a healthy one, and then, damaged mitochondria would be enclosed by autophagosomes and further degraded in the autolysosomes [44]. Remarkably, endurance training exercises could control the content of PINK1, Parkin, MFN1, and MFN2 for alleviating the mitochondria in MAFLD [45]. The ubiquitination of mitochondrial outer membrane proteins such as voltage-dependent anion channel 1 (VDAC1) will promote the recognition of VDAC1 by autophagy receptors such as histone deacetylase 6 (HDAC6) [46]. VDAC1 is an OMM protein adjusting the energy homeostasis as a mitochondrial gatekeeper, which is involved in mitochondria-mediated apoptosis [47]. Protecting mitochondria by inhibiting the oligomerization of VDAC1 might reduce the injury of hepatocytes in MAFLD [48]. PINK1 might also be slashed by protease PARL in the mitochondrial matrix, and then, the shortened form of PINK1 might be further degraded by the ubiquitin proteasome in cytoplasm [49]. PINK1 located in OMM could employ autophagy receptors including nuclear dot protein 52 (NDP52), p62, and optineurin (OPTN), which could attach to light-chain 3 (LC3) to conjoin the damaged mitochondria and phagosomes, and lastly, damaged mitochondria would be despoiled in autolysosome [50]. To further activate mitophagy, PINK1 could again phosphorylate the ubiquitin-like domain of Parkin to facilitate Parkin localization from the cytosol to the OMM of damaged mitochondria [51]. 

An immunosuppressive drug rapamycin, one of mechanistic/mammalian target of rapamycin (mTOR) inhibitors, has been shown to prevent the development of hepatic steatosis [52], suggesting that mTOR could influence the pathology of MAFLD by regulating the inflammatory response [53]. The rapamycin could also induce autophagy by inhibiting the mTOR signaling pathway [54] (Figure 2). In general, the mTOR signaling may play a vital role in the proliferation and/or differentiation of various cells. Remarkably, the downmodulation of mTOR could increase the longevity of cells [55]. Metformin can downregulate the phosphorylation of S6 kinase, which is an important signaling molecule downstream of mTOR. Additionally, metformin can be used to treat a diversity of aging-related disorders, such as cardiovascular disease or cognitive decline [56]. The mTOR is involved in two protein complexes with diverse functions [57]. However, the mTOR complex 1 (mTORC1) may impede autophagy by integrating the upstream signals with PI3K and/or AKT. In addition, the mTOR complex 2 (mTORC2) may not be closed with the regulation of autophagy [58] (Figure 2). The activated PI3K could trigger the activation of AKT via PDK1 [59]. Then, the activated AKT further phosphorylates the tuberous sclerosis protein 2 (TSC2) and blocks its interplay with TSC1, and eventually results in the activation of mTORC1 [60]. As an energy sensor, adenosine monophosphate (AMP)-activated protein kinase (AMPK) could also regulate the mTORC1 [61]. The AMPK might be stimulated by a decline in ATP concentration which increases the ratio of AMP/ATP [62]. Interestingly, the longevity-enhancing effects of the metformin may imitate the effect of the stimulation of AMPK [63]. Therefore, AMPK-mTOR signaling pathways might regulate the various characteristics of cellular homeostasis including cell proliferation, differentiation, energy metabolism, and autophagy [64]. The modulation of autophagy has been revealed as a probable therapeutic target in MAFLD [65]. Interestingly, it has been reported that prebiotic fiber intake could ameliorate the hepatic pathology via the modulation of AMPK activity [66]. In addition, prebiotic fructo-oligosaccharides could promote the tight junction assembly of intestinal epithelium through AMPK activation, which may contribute to the treatment of diseases with impaired intestinal barrier function [67]. The dysfunction and dysregulation of the intestinal barrier may impair mucosal immune tolerance, leading to an increased risk of developing metabolic disorders, particularly MAFLD [68].

## 4. Valuable Components from Natural Foods for the Treatment of MAFLD

Cumulative evidence highlights a favorable role of the naturally derived compounds from various natural sources for the prevention and/or treatment of metabolic disorders including MAFLD [69]. Currently, several composites with a liver-protective effect may be a potential alternative therapy for the complications of MAFLD due to their unique therapeutic properties and considerable safety [70]. Additionally, there are a variety of natural compounds with known autophagy-modulating properties that also control the production of ROS [71,72]. In particular, the potential of targeting mitophagy with phytochemicals for the possible management of MAFLD has been shown here, with a view providing a direction for finding some phytochemicals that target mitophagy to prevent and/or treat MAFLD [73]. Prebiotics enhance the secretion of GLP-1, in part via the upregulation of the proglucagon gene expression in the distal intestine [74]. Furthermore, the fermentation of prebiotics leads to the production of SCFAs [75]. For example, prebiotic fibers could promote the growth of *Bifidobacterium* spp. in the gut, which has been shown to reduce the metabolic endotoxemia associated with the consumption of a high-fat diet [76]. In addition, inulin fiber has been used as a prebiotic to alleviate glucose and/or lipid metabolism disorders by modulating the gut microbiota [77].

Resveratrol is a natural polyphenolic compound, which may show several helpful effects on the hepatocyte of MAFLD [78] through the possible activation of AMPK signaling [79]. In general, dietary polyphenols may exhibit antioxidative, anti-inflammatory, and anticancer properties, which may lead to the restoration of the otherwise-impaired inflammatory diseases. Resveratrol could decrease the mTORC1 signaling for the stimulation of autophagy [80] (Figure 2). In addition, resveratrol could also encourage autophagy in embryonic stem cells, which may also be delivered by the associated suppression of the mTORC1 [81]. A prebiotic resveratrol could be a potential candidate for the treatment of obesity and MAFLD [82]. Anthocyanins are common plant pigments, which could also stimulate autophagy [83] with potential strong activity against MAFLD [84]. Anthocyanins could induce autophagy via the AMPK-mTOR pathway [85]. The positive effects of anthocyanins on the accumulation of hepatic lipid might be mediated via the activation of autophagy, suggesting a capability for the therapeutic tactics in MAFLD [86]. In fact, it has been shown that resistant starch, including anthocyanins, could ameliorate insulin resistance, dyslipidemia and liver injury [87]. Trehalose is a natural saccharide that could alter autophagy [88], initiating lysosomal temporary damage [89]. A health benefit for trehalose has been proposed in HepG2 cells [90]. The action mechanisms of trehalose may involve the inhibition of glucose transporters leading to AMPK-activation, which could affect the autophagy [91]. Thus, the trehalose could work as a fragile inhibitor of the lysosome via the inhibition of mTORC1 [92]. In addition, trehalose supplementation could act as a prebiotic to selectively promote the growth of beneficial bacteria [93], which may attenuate hepatic steatosis [94]. 

The aforementioned phytochemicals may have the capability to target putative molecular and/or biochemical actions in mitophagy. In view of the fact that more and more various phytochemicals would be applied to the treatment of MAFLD, it is necessary to have a widespread understanding of the effects and/or potential mechanisms of phytochemicals on MAFLD [95]. In general, phytochemicals may be a pleiotropic molecule with the capability to interact with the different cellular targets involved in inflammation conditions [96]. For example, it has been revealed that genistein could also downregulate the NF-κB signaling pathway, leading to a decrease in the expression of IL-1 and/or IL-6 [97]. On the other hand, genistein could also down-regulate Bcl-2 and up-regulates Bax, which might suppress cell growth and induces apoptosis [98]. There is a need to further explore the underlying mechanisms. More research should focus on the regulatory mechanisms of mitophagy in MAFLD, and further search for the potential of targeting mitophagy with certain phytochemicals for the prevention and/or treatment of MAFLD [99]. The use of such compounds could be an inaugural point for the new therapy of MAFLD.

## 5. Improved Mitophagy with the Modification of Gut Microbiota for the Treatment of MAFLD

Autophagic dysfunction may contribute to hepatic steatosis and accelerate the progression of MAFLD [100]. Therefore, mitophagy activation might play a key role in improving the situation of MAFLD. In addition, mitophagy may account for the mitochondrial degradation occurring in the presence of stressful conditions, which is detected in several disorders such as Parkinson’s disease [101]. The mitochondria dysfunctions suppress the β-oxidation of mitochondria and increase the production of the toxic lipid metabolism, which further adversely affects the mitochondrial dysfunction [102]. Interestingly, the gut microbiota could produce short-chain fatty acids (SCFAs), which might act as signal molecules to regulate the autophagy [103,104]. Therefore, the modulation of microbiota has been shown to control the machinery of autophagy against liver toxicity [105]. The gut microbiota may regulate the autophagy progression through multiple mechanisms with regulating cellular metabolisms [106] and/or epigenetically inhibiting histone deacetylases [107]. SCFAs produced by the microbiota can be found in peripheral blood, where they are taken up by organs, as they act as signal molecules [103]. Therefore, it is reasonable to consider that gut dysbiosis, resulting in the dysregulation of SCFA production, increases the susceptibility of liver injury. In fact, MAFLD pathogenesis is closely associated with gut dysbiosis [108]. The gut dysbiosis could accelerate the development of MAFLD due to abnormal levels of gut microbial metabolites and/or their effects on gut epithelial permeability [109,110]. Therefore, it has been shown that the gut microbiota might play an imperative role in the progression of MAFLD, and gut microbiota dysbiosis may be a significant factor in MAFLD pathogenesis [111,112].

Many factors could affect the composition of gut microbiota, including the host age, sex, menopausal status, host immunity, exposure to antibiotics, and dietary behaviors [113]. Specific dietary factors and/or the presence of specific bioactive compounds may affect the diversity of gut microbiota [114]. In addition, insoluble fiber, fat, and protein contents may have important effects on the structure of gut microbiota [115]. Metabolic diseases such as obesity and diabetes are also related with the gut microbiota [116]. Therefore, restoring the gut microbiota with commensal bacteria may help improve the pathology [117]. Interestingly, it has been shown that the amount of *Bacteroidetes* may decrease in MAFLD, while those of *Firmicutes* and *Proteobacteria* may be increased [118]. Gut microbiota has been shown to regulate key transcription factors, and enzymes related to mitochondrial biogenesis and metabolism. In addition, microbiota metabolites seem to directly affect mitochondrial oxidative stress and the formation of mitophagy, thereby regulating the activation and/or production of inflammatory cytokines. Furthermore, liver is exposed to endotoxins such as lipopolysaccharide (LPSs), which can be delivered to the liver or endogenously produced by gut microbiota [119]. Yeast Fermentate Prebiotics may have beneficial roles in maintaining the health of the host [120], which could be associated with balancing the gut microbial community, regulating immunity [121]. In particular, prebiotics are used by microorganisms as food, which could exert a beneficial effect on the health of the host. Currently available prebiotics include lactulose, fibers, inulin derivatives, and milk oligosaccharides. In addition, phytochemicals may act as prebiotics in the gut lumen for autophagy inducers [122]. 

## 6. Future Perspectives

Multiple risk factors including obesity could influence the development and progression of MAFLD [123]. It may be indispensable to explore the roles of mitochondrial ROS and mitophagy in MAFLD. Interestingly, ferulic acid could have prebiotic effects on the inflammatory responses of the host through the activation of the Nrf2 signaling pathway, which could decrease the concentration of ROS [124]. In general, mitophagy could selectively degrade impaired mitochondria to suppress the impaired mitochondria-derived ROS that would injure healthy mitochondria, resulting in mitochondrial dysfunction. As a major mechanism of mitochondrial quality control, therefore, mitophagy could also degrade dysfunctional mitochondria to maintain mitochondrial integrity in MAFLD [125]. Consequently, activated mitophagy would eventually prevent ROS-triggering oxidative stresses and/or inflammatory responses. MAFLD may be closely linked to an unhealthy lifestyle. In particular, the Western diet considered by excessive energy intake, the frequent consumption of meat, processed or artificial foods, sugar-sweetened drinks, and unhealthy ways of cooking, may lead to the development of MAFLD [126]. Looking for natural compounds with appropriate mitophagic activities could offer new intuitions into the therapeutic mediation for impaired mitochondria-related diseases including MAFLD. However, the therapeutic potential of autophagy alteration has not been entirely exploited. In addition, a synbiotic combination of probiotic and prebiotic agents could efficiently affect the composition of the fecal microbiome in patients with MAFLD [127]. The changes involved in hepatic inflammation may support a requisite to evaluate the effect of the synbiotic treatment of MAFLD [127]. Therefore, additional efforts should be directed towards cultivating novel compounds with improved potency and safety. Detailed mechanistic insights are also required to clarify the precise function of autophagy in the pathogenesis of MAFLD. It has been suggested that the PI3K/AKT/mTOR signaling pathway might be involved in the regulation of Th17/Treg balance, which might contribute to the pathogenesis of MAFLD [128] (Figure 3). In addition, the Th17/Treg ratio in the liver might be valuable for classifying patients with MAFLD/NASH from those with simple or a light degree of steatosis [128]. The Th17/Treg balance has been also suggested to play an important role in the patho-physiology of major depressive disorders [128]. Now, we are confident that other intricate diseases including neurodegenerative diseases and/or aging-related disorders may also play a role in the mechanistic pathogenesis of MAFLD. Therefore, a prevalent inclusion of natural molecules in functional foods deserves consideration, as it could vitally contribute to the prevention/treatment of MAFLD as well as widely apply to the public health domain. Furthermore, no curative therapeutic approach for such diseases has been found to date, so extensive studies are required so that we may one day find remedies to these lethal diseases.

## Figures and Tables

**Figure 1 ijms-24-05465-f001:**
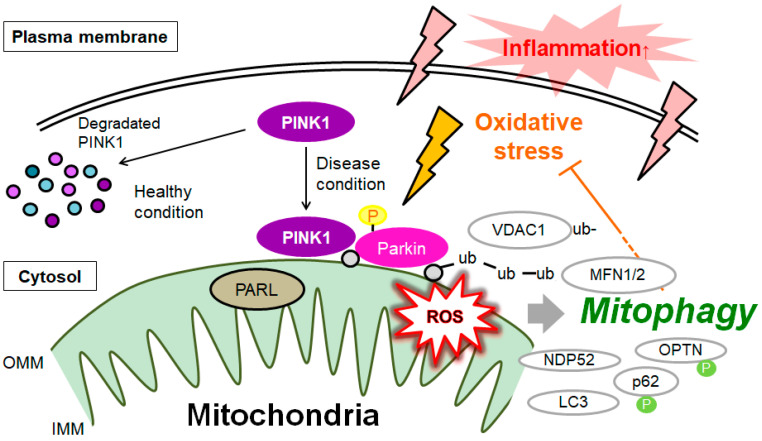
A theoretical graphic representation and overview of PINK1, Parkin, and related molecules in regulatory pathway for mitophagy. Under the healthy and steady state of cells, PINK1 is degraded within the surface of the mitochondria. This may be subdued by mitochondrial damage due to oxidative stress, resulting in PINK1 and Parkin accumulation in the outer membrane of mitochondria. PINK1 could phosphorylate ubiquitin to activate Parkin ubiquitin ligase activity. In addition, the Parkin is also assumed to be phosphorylated and ubiquitinated, resulting in the induction of mitophagy. Note that some critical pathways have been omitted for clarity. OMM, outer mitochondrial membrane; IMM, inner mitochondrial membrane; MARK2, microtubule affinity regulating kinase 2; MFN1, mitofusin 1; MFN2, mitofusin 2; NDP52, nuclear dot protein 52; PARL, presenilin-associated rhomboid-like; OPTN, optineurin; PINK1, PTEN-induced kinase 1; VDAC1, voltage-dependent anion channel 1; Ub, ubiquitin.

**Figure 2 ijms-24-05465-f002:**
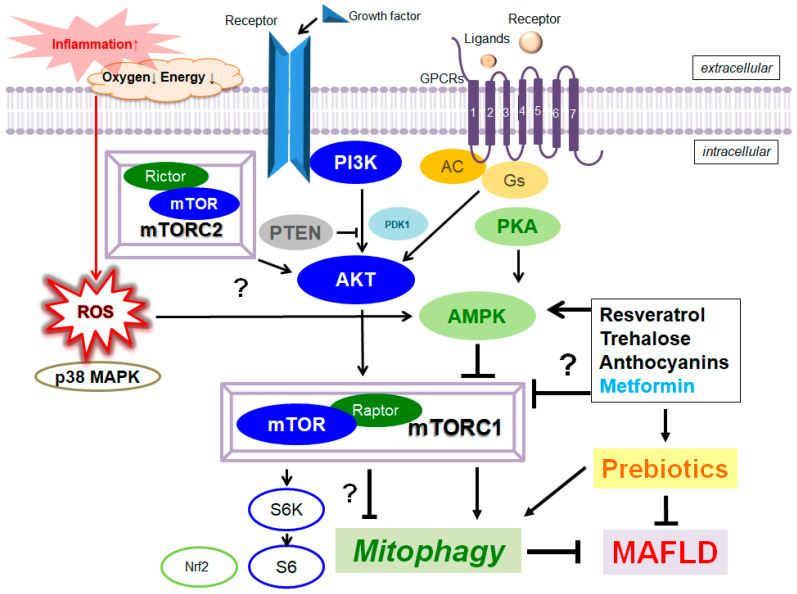
Several modulator molecules linked to the PI3K/AKT/mTOR/AMPK/mTORC1/ mTORC2 signaling pathway are demonstrated. Example compounds known to act on AMPK/mTOR and/or mitophagy/autophagy signaling are also shown on the right side. The arrowhead means stimulation, whereas hammerhead represents inhibition. Note that some critical events such as immune activation and/or antioxidant feedback have been omitted for clarity. Abbreviation: AMPK, adenosine monophosphate-activated protein kinase; mTOR, mammalian/mechanistic target of rapamycin; PI3K, phosphoinositide-3 kinase; PKA, protein kinase A; PTEN, phosphatase and tensin homologue deleted on chromosome 10; mTORC, mechanistic/mammalian target of rapamycin complex; MAFLD, metabolic dysfunction-associated fatty liver disease; mTOR, mechanistic/mammalian target of rapamycin; mTORC 1/2, mechanistic/mammalian target of rapamycin complex 1/2.

**Figure 3 ijms-24-05465-f003:**
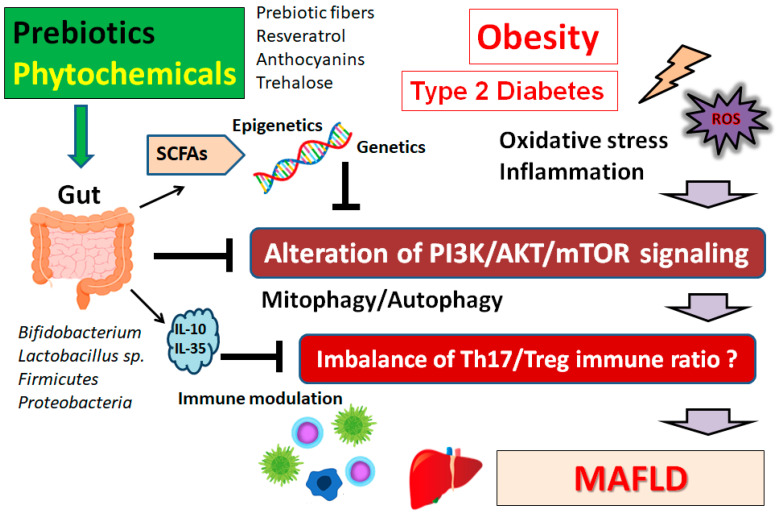
Schematic representation of the inhibition against the pathogenesis of MAFLD. Some kinds of prebiotics and/or phytochemicals could contribute to the alteration of the gut microbial community and play a valuable role in modifying the PI3K/AKT/mTOR signaling pathway, partly via the epigenetical regulation with SCFAs. Examples of certain beneficial microbial species with several effects on MAFLD have been shown at the left side. The arrowhead indicates stimulation whereas hammerhead shows inhibition. Note that several important activities such as cytokine-induction and/or inflammatory reactions have been omitted for clarity. Abbreviation: MAFLD, metabolic-associated fatty liver disease; mTOR, mammalian/mechanistic target of rapamycin; PI3K, phosphoinositide-3 kinase; ROS, reactive oxygen species; SCFAs, short-chain fatty acids.

## Data Availability

Not applicable.

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
