# Peer review of "Tactics with Prebiotics for the Treatment of Metabolic Dysfunction-Associated Fatty Liver Disease via the Improvement of Mitophagy"

_ijms, 2023, doi:10.3390/ijms24065465_

Round 1

Reviewer 1 Report

In the manuscript entitled “Tactics with prebiotics for the treatment of metabolic dysfunction associated fatty liver disease via the improvement of mitophagy”, Ai Tsuji et al. reviewed that mitophagy plays a protective role in MAFLD and discussed the molecular mechanism of mitochondrial dysfunction in MAFLD and discuss the potential therapeutics with prebiotics targeting autophagy for the treatment of MAFLD, then they discussed the potential of prebiotics to treat MAFLD. The manuscript is well-written, and the topic is very intriguing.

1.     In the first paragraph, the completion “It has been shown that pectin could protect the liver from metabolic injuries caused by alcohol and/or fatty diet” should have a citation.

2.     In the second paragraph, the authors mentioned that NAFLD has become among the top leading indications for liver transplantation. The conclusion is not accurate.

3.     The authors should explain the full name when referring to NAFLD for the first time.

4.     Language needs to be improved throughout the manuscript. Some grammatical errors and typos are noted. For example, in the first paragraph, “Furtheremore” and “benn”

5.     In parts 2 and part 3, the authors should discuss mitochondrial dysfunction and the molecular mechanism more logically.

6.     If possible, please discuss more polyphenolic compounds that can act as prebiotics.

Author Response

Reviewer1

In the manuscript entitled “Tactics with prebiotics for the treatment of metabolic dysfunction associated fatty liver disease via the improvement of mitophagy”, Ai Tsuji et al. reviewed that mitophagy plays a protective role in MAFLD and discussed the molecular mechanism of mitochondrial dysfunction in MAFLD and discuss the potential therapeutics with prebiotics targeting autophagy for the treatment of MAFLD, then they discussed the potential of prebiotics to treat MAFLD. The manuscript is well-written, and the topic is very intriguing.

Thank you so much for the good evaluation to our manuscript.

  1. In the first paragraph, the completion “It has been shown that pectin could protect the liver from metabolic injuries caused by alcohol and/or fatty diet” should have a citation.

We added the citation at the end of this sentence. Ref 6 is a citation for the completion.

  1. In the second paragraph, the authors mentioned that NAFLD has become among the top leading indications for liver transplantation. The conclusion is not accurate.

According to the suggestion, we have amended the text of the second paragraph.

  1. The authors should explain the full name when referring to NAFLD for the first time.

According to the suggestion, nonalcoholic fatty liver disease (NAFLD) has been mentioned at the first place appeared.

  1. Language needs to be improved throughout the manuscript. Some grammatical errors and typos are noted. For example, in the first paragraph, “Furtheremore” and “benn”

According to the suggestion, again we have gone over the text/abstract and amended typos and grammatical errors as much as possible to improve the manuscript more helpful to the readers. In addition, the “Furtheremore” and “benn” have been replaced with the “Furthermore” and “been”, respectively.

  1. In parts 2 and part 3, the authors should discuss mitochondrial dysfunction and the molecular mechanism more logically.
  2. If possible, please discuss more polyphenolic compounds that can act as prebiotics.

We have added some explanation about the polyphenolic action.

Reviewer 2 Report

In this article, Tsuji, Matsuda, and their colleagues applied the depth of knowledge built in other multifactorial disorders, such as aging-related disorders, to MAFLD and discussed potential preventive/therapeutic implications of functional foods and existing medications to modulate autophagy/mitophagy and reverse dyshomeostasis observed in MAFLD. I read this review with great interest and agree that there are common pathways between MAFLD and other chronic human diseases involving cellular dyshomeostasis in different organs, which could have therapeutic implications for MAFLD. 

However, most of this review was spent on summarizing the mechanistic importance of autophagy and mitophagy in MAFLD, which is well-known in this field. The authors can improve the contribution of this review by focusing more on what we know and don't know about these compounds in their application to human MAFLD. 

MAFLD is a multifactorial disorder driven by metabolic derangements induced by obesity. MAFLD pathogenesis is multiphasic and complex. Thus, targeting a single pathway is unlikely to be effective in most patients with MAFLD. The pleiotropic nature of functional foods, favorable safety profile, and low cost may be suitable for preventing/treating this highly prevalent disease globally. One of the challenges in clinical MAFLD is its heterogeneity. Knowing which subgroup of patients should be targeted by a specific therapeutic approach is crucial. Although they discussed the depth of mechanistic understanding around autophagy regulation and mitochondrial homeostasis, they did not discuss biological disparities in these pathways. I want the authors to consider such critical gaps and expand their discussion, including biological disparities, hinting at potential target populations for specific compounds. Please note, sex differences are well-reported in mitochondrial functions and autophagy. This reviewer fully acknowledges that the concept of biological variations by age, sex, and reproductive status is not yet fully appreciated in the basic sciences. However, successful translation from experiments to humans requires such knowledge. If no data are available, the authors should acknowledge this as a critical knowledge gap for translating into humans in future perspectives. There are some minor points for the authors' consideration.

  1. MAFLD is 'metabolic-associated fatty liver disease'. Please be consistent and use the established term. MAFLD was introduced to reclassify fatty liver disease driven by obesity-related metabolic derangement as a single entity. However, it does not exclude other causes, such as alcoholic liver disease, chronic hepatitis B or C, or autoimmunity. Thus, MAFLD is also heterogeneous. It is based on the clinical belief that metabolic derangement drives the disease progression without excluding other superimposing causes, accurately reflecting real-world patients. For this review, using MAFLD, as opposed to NAFLD, is appropriate, but the authors need to acknowledge the complexity of the clinical definition of MAFLD.
  2. There are a few areas where they mentioned "neurons" instead of hepatocytes (Page 2, Page 3). These should be corrected.
  3. Overexpression of ROS (Page 3) should be stated as overproduction of ROS.
  4. Many factors contribute to the overproduction of ROS. It is not just inflammation. In MAFLD, increased energy influx to the liver (e.g., FFA) is a primary factor driving ROS generation (Page 3, under Dysfunction of mitochondria involved in the pathogenesis of MAFLD.
  5. "Increase of damaged mitochondria and/or overproduction of ROS would fortify each other (Page 3)." Did the authors mean an increase of damaged mitochondria AND overproduction of ROS would "augment" each other, forming a vicious cycle?  
  6. "Therefore, mitochondrial accurate quality control should be performed for preventing liver diseases (Page 3)." Did the authors mean, "Mitochondrial quality control is crucial to preventing disease progression of liver diseases"? They might want to rephrase this.
  7. Should mitochondrial synthesis (Page 3) be expressed as mitochondrial "biogenesis"? 
  8. "Many factors could affect the composition of gut microbiota, including the host age, host immunity, exposure to antibiotics, and dietary behaviors" (page 7). – Please note that gut microbiota also differs by sex and menopausal status. 
  9. In Sections 3 and 4, subtitles for different compounds would be helpful before describing the molecules, sources, mechanisms, and potential implications in human MAFLD.

Author Response

Reviewer2

In this article, Tsuji, Matsuda, and their colleagues applied the depth of knowledge built in other multifactorial disorders, such as aging-related disorders, to MAFLD and discussed potential preventive/therapeutic implications of functional foods and existing medications to modulate autophagy/mitophagy and reverse dyshomeostasis observed in MAFLD. I read this review with great interest and agree that there are common pathways between MAFLD and other chronic human diseases involving cellular dyshomeostasis in different organs, which could have therapeutic implications for MAFLD. 

However, most of this review was spent on summarizing the mechanistic importance of autophagy and mitophagy in MAFLD, which is well-known in this field. The authors can improve the contribution of this review by focusing more on what we know and don't know about these compounds in their application to human MAFLD. 

MAFLD is a multifactorial disorder driven by metabolic derangements induced by obesity. MAFLD pathogenesis is multiphasic and complex. Thus, targeting a single pathway is unlikely to be effective in most patients with MAFLD. The pleiotropic nature of functional foods, favorable safety profile, and low cost may be suitable for preventing/treating this highly prevalent disease globally. One of the challenges in clinical MAFLD is its heterogeneity. Knowing which subgroup of patients should be targeted by a specific therapeutic approach is crucial. Although they discussed the depth of mechanistic understanding around autophagy regulation and mitochondrial homeostasis, they did not discuss biological disparities in these pathways. I want the authors to consider such critical gaps and expand their discussion, including biological disparities, hinting at potential target populations for specific compounds. Please note, sex differences are well-reported in mitochondrial functions and autophagy. This reviewer fully acknowledges that the concept of biological variations by age, sex, and reproductive status is not yet fully appreciated in the basic sciences. However, successful translation from experiments to humans requires such knowledge. If no data are available, the authors should acknowledge this as a critical knowledge gap for translating into humans in future perspectives. There are some minor points for the authors' consideration.

Thank you so much for the good evaluation to our manuscript. However, it is too difficult for us at this stage to revise the manuscript with the discussion of biological disparities and sex differences in mitochondrial functions and autophagy. We would like to describe them in the future opportunity.

  1. MAFLD is 'metabolic-associated fatty liver disease'. Please be consistent and use the established term. MAFLD was introduced to reclassify fatty liver disease driven by obesity-related metabolic derangement as a single entity. However, it does not exclude other causes, such as alcoholic liver disease, chronic hepatitis B or C, or autoimmunity. Thus, MAFLD is also heterogeneous. It is based on the clinical belief that metabolic derangement drives the disease progression without excluding other superimposing causes, accurately reflecting real-world patients. For this review, using MAFLD, as opposed to NAFLD, is appropriate, but the authors need to acknowledge the complexity of the clinical definition of MAFLD.

Thank you very much. You may be an excellent and brilliant clinician, while we are not but basic scientists. We keep in mind it. The “metabolic-associated fatty liver disease” has been replaced for the explanation of MAFLD at the first place it appeared.

  1. There are a few areas where they mentioned "neurons" instead of hepatocytes (Page 2, Page 3). These should be corrected.

Thank you so much. These have been corrected.

  1. Overexpression of ROS (Page 3) should be stated as overproduction of ROS.

Thank you so much. The sentence has been corrected, accordingly.

  1. Many factors contribute to the overproduction of ROS. It is not just inflammation. In MAFLD, increased energy influx to the liver (e.g., FFA) is a primary factor driving ROS generation (Page 3, under Dysfunction of mitochondria involved in the pathogenesis of MAFLD.

Yes, of course. Accordingly, we have added the following sentence, “In addition, high-fat and high-cholesterol diet could greatly raise the blood ROS level, suggesting that increased energy influx to the liver might be involved in the ROS generation [21].”

  1. "Increase of damaged mitochondria and/or overproduction of ROS would fortify each other (Page 3)." Did the authors mean an increase of damaged mitochondria AND overproduction of ROS would "augment" each other, forming a vicious cycle?  

Yes, that is. The verb "augment" has been replaced with the "fortify", adding the “forming a malicious cycle”.

  1. "Therefore, mitochondrial accurate quality control should be performed for preventing liver diseases (Page 3)." Did the authors mean, "Mitochondrial quality control is crucial to preventing disease progression of liver diseases"? They might want to rephrase this.

Thank you very much. Yes, that is fine.

  1. Should mitochondrial synthesis (Page 3) be expressed as mitochondrial "biogenesis"? 

True, the "biogenesis" might be better.

  1. "Many factors could affect the composition of gut microbiota, including the host age, host immunity, exposure to antibiotics, and dietary behaviors" (page 7). – Please note that gut microbiota also differs by sex and menopausal status. 

The term “sex and menopausal status” has been inserted in the sentence, as “Many factors could affect the composition of gut microbiota, including the host age, sex and menopausal status, host immunity, and exposure to antibiotics, and dietary behaviors.”

  1. In Sections 3 and 4, subtitles for different compounds would be helpful before describing the molecules, sources, mechanisms, and potential implications in human MAFLD.

Adding the subtitles, we would have to describe something contents there, which may be further well-known contents in this field. Besides, there are few real evidence for the potential implications of natural food compounds or prebiotics in human MAFLD.

Reviewer 3 Report

Tsuji and colleagues discuss the role mitochondrial dysfunction in „metabolic dysfunction-associated fatty liver disease (MAFLD), a disease which is strongly influenced in their development and progression by deskbound lifestyles, dietary changes, epidemic obesity and type 2 diabetes. The authors describe possible molecular mechanism of mitochondrial dysfunction including the role of mitophagy in MAFLD and discuss potential therapeutics with prebiotics or components from natural food respective several phytochemicals for the treatment of MAFLD.

The review is written clearly and understandably. Nevertheless, there are a few critical remarks as indicated.

The authors refer to hepatocytes that represent the main cell type of liver. Still, there is a liver-specific mesenchymal cell type, who is known to be involved into the progression of fibrotic altered liver, which can result in steatosis and NASH: the hepatic stellate cell. In addition, hepatic cell types taken together as such stellate cells, endothelial cells, Kupffer cells and hepatocytes interact with each other. This aspect should be considered or discussed since autophagy/mitophagy could be also found in these cells. Thus other hepatic cell types can participate in the progression of MAFLD!? Is known if tactics with prebiotics for the treatment of MAFLD are associated with other hepatic cell types?

Minor

In Figure 2, some of the labeled boxes are not legible. Separate the groups of phytochemicals and other components (metformin). Please more clearly in the use of terms stimulation and inhibition. It is not yet clear considering resveratrol, trehalose, anthocyanins, metformin.

Figure 3, which cells are involved in the immunomodulation?

The numbers in brackets have different size, e.g. page 8.

Formatting of the references is inconsistent, e.g. page 12, 13, 17, 18.

Author Response

Reviewer3

Tsuji and colleagues discuss the role mitochondrial dysfunction in „metabolic dysfunction-associated fatty liver disease (MAFLD), a disease which is strongly influenced in their development and progression by deskbound lifestyles, dietary changes, epidemic obesity and type 2 diabetes. The authors describe possible molecular mechanism of mitochondrial dysfunction including the role of mitophagy in MAFLD and discuss potential therapeutics with prebiotics or components from natural food respective several phytochemicals for the treatment of MAFLD.

The review is written clearly and understandably. Nevertheless, there are a few critical remarks as indicated.

Thank you so much for the good evaluation to our manuscript.

The authors refer to hepatocytes that represent the main cell type of liver. Still, there is a liver-specific mesenchymal cell type, who is known to be involved into the progression of fibrotic altered liver, which can result in steatosis and NASH: the hepatic stellate cell. In addition, hepatic cell types taken together as such stellate cells, endothelial cells, Kupffer cells and hepatocytes interact with each other. This aspect should be considered or discussed since autophagy/mitophagy could be also found in these cells. Thus other hepatic cell types can participate in the progression of MAFLD!? Is known if tactics with prebiotics for the treatment of MAFLD are associated with other hepatic cell types?

True, we think that other hepatic cell types could participate in the progression of MAFLD. As shown that we have cited several references suggesting the effectiveness of prebiotics even in neuronal cells, we think the tactics with prebiotics might be also effective for the other hepatic cell types.

Minor

In Figure 2, some of the labeled boxes are not legible. Separate the groups of phytochemicals and other components (metformin). Please more clearly in the use of terms stimulation and inhibition. It is not yet clear considering resveratrol, trehalose, anthocyanins, metformin.

There are many controverting and/or contradictory reports showing effects of these compounds whether they stimulate or inhibit. We cannot make it clearer at present. Additionally, figures have been improved a little for clarity.

Figure 3, which cells are involved in the immunomodulation?

At present also, it is not clear to be involved in the immunomodulation. We guess it might be glial cells in brain that might be involved in the immunomodulation. However, what bacteria in gut could alter the function of glial cells in brain has not been revealed.

The numbers in brackets have different size, e.g. page 8.

Sorry, the numbers in brackets are all equal size in our draft manuscript. That could be altered at the editorial level.

Formatting of the references is inconsistent, e.g. page 12, 13, 17, 18.

Thank you so much. We have improved the format of references.

Round 2

Reviewer 3 Report

The authors have inserted the corrections accordingly. I have only a few minor comments. The numbers for the citations have a different size (e.g. page 8 line 317) throughout the text. The names for the bacteria also have a large font size (e.g. page 6, line 255; page 8, lines 325-326).

Author Response

All the numbers for the citations as well as the names for the bacteria in the text have been improved to be equally font-sized. Thank you so much.